# Association between Dietary Patterns and Chronic Obstructive Pulmonary Disease in Korean Adults: The Korean Genome and Epidemiology Study

**DOI:** 10.3390/nu13124348

**Published:** 2021-12-02

**Authors:** Moon-Kyung Shin, Se Hyun Kwak, Youngmok Park, Ji Ye Jung, Young Sam Kim, Young Ae Kang

**Affiliations:** 1Division of Pulmonary and Critical Care Medicine, Department of Internal Medicine, Severance Hospital, Yonsei University College of Medicine, 50-1 Yonsei-ro, Seodaemun-gu, Seoul 03722, Korea; tyu0712@naver.com (M.-K.S.); 0MOKFV@yuhs.ac (Y.P.); STOPYES@yuhs.ac (J.Y.J.); YSAMKIM@yuhs.ac (Y.S.K.); 2Division of Pulmonology, Allergy and Critical Care Medicine, Department of Internal Medicine, Yongin Severance Hospital, Yonsei University College of Medicine, 363 Dongbaekjukjeon-daero, Giheung-gu, Yongin-si 16995, Korea; sehyun31@yuhs.ac; 3Institute of Immunology and Immunological Diseases, Yonsei University College of Medicine, 50-1 Yonsei-ro, Seodaemun-gu, Seoul 03722, Korea

**Keywords:** chronic obstructive pulmonary disease, dietary pattern, lung function, risk factors

## Abstract

In addition to smoking, dietary habits may contribute to the development of chronic obstructive pulmonary disease (COPD). This study aimed to examine the association between dietary patterns and lung function in a Korean community cohort. A total of 5436 participants were included from the Ansan–Ansung cohort study. To identify the dietary patterns, we performed principal component factor analysis using the results of a semi-quantitative food frequency questionnaire. The forced expiratory volume in 1 s (FEV1), forced vital capacity (FVC), and FEV1/FVC ratio were measured by spirometry. Multiple logistic regression models were used to evaluate the association between dietary patterns and lung function after adjusting for confounders. We identified four major dietary patterns; ‘prudent’, ‘coffee, fat, and sweet’, ‘westernized’, and ‘white rice’. After adjusting for potential confounders, the ‘coffee, fat, and sweet’ dietary pattern was negatively associated with lung function, particularly the FEV1/FVC ratio. Participants with high scores for the ‘coffee, fat and sweet’ pattern had a higher risk of COPD among men but not women. Therefore, these results indicate that the ‘coffee, fat and sweet’ dietary pattern is inversely related to lung function in Korean adults. Our results indicate that dietary habits may be modifiable risk factors for COPD.

## 1. Introduction

Chronic obstructive pulmonary disease (COPD) is characterized by progressive airflow obstruction [1,2,3] and is currently the third leading cause of chronic morbidity and mortality worldwide, causing 3.23 million deaths in 2019 [4,5].

Although tobacco smoking is known to be the most common risk factor for COPD, from 25% to 45% of patients with airway obstruction are nonsmokers [6,7]. As nonsmokers account for a significant proportion of COPD patients, a deeper understanding of the other risk factors for COPD such as environmental exposure, occupational hazards, and dietary habits, becomes more important. While the relationship between lifestyle factors and the risk of developing other chronic diseases with similar burdens on quality of life, such as cancer, cardiovascular disease, or diabetes, has been reported in several previous studies [8,9,10,11], little is known about the relationship between lifestyle factors and the risk of COPD.

Dietary intake may be an important risk factor for impaired lung function, and healthy dietary habits may be protective against COPD [12]. In the past, many studies have focused on the effects of individual foods or nutrients [13]. An approach that analyzes dietary patterns rather than the individual constituents of food has recently become more appealing [14]. Such a pattern analysis may reflect the dietary habits more accurately as foods are consumed together, with interactions between individual nutrients. People eat meals with complex combinations of nutrients that are likely to be interactive or synergistic [15]. A single nutrient analysis might be confounded by the effect of dietary patterns [15,16]. In addition, considering how foods and nutrients are consumed in combination would more closely reflect the real world. Thus, a dietary pattern analysis in which multiple dietary components are operationalized as a single exposure could provide new insights into the association of diet and COPD [17,18].

However, there is still a lack of epidemiological studies on dietary patterns, and the impact of dietary intake on the risk of developing COPD. In this study, we aimed to determine the dietary patterns by using principal components analysis, and to investigate the association between dietary patterns and COPD. Additionally, we investigated the association of dietary patterns and their spirometric indices.

## 2. Methods

### 2.1. Study Population

The current study was based on the Korean Genome and Epidemiology Study (KoGES) [19], a large consortium project consisting of six population-based prospective cohort studies. The datasets were obtained from the Ansan and Ansung study among the six cohort studies. The Ansan and Ansung study recruited Korean adults who were aged from 40 to 69 years at the baseline from 2001 to 2002 [20]. The total number of participants in the base survey was 10,030, and the 7th follow-up survey and examinations were completed by 2016. All participants provided informed consent for the baseline data and biospecimens and underwent an interview and physical examination. The participants were questioned by trained interviewers regarding their socio-demographic status, lifestyle along with anthropometric measurements, and diet [19].

Participants for the current study were recruited from the 3rd follow-up assessment between 2005 and 2006. Among the 6231 eligible individuals, participants without more than two valid lung function measurements (*n* = 368), more than two bio-electrical impedance analyses (InBody 3.0, Biospace, Seoul, Korea) measurements [21] (*n* = 395), smoking history (*n* = 4), and implausible energy intake (<500 or >5000 kcal/day, *n* = 28) were excluded. As a result, a total of 5436 participants were included in the final analysis (Figure 1).

### 2.2. Dietary Assessment

Certified clinical dietitians used a semi-quantitative food frequency questionnaire (SQFFQ) to assess the dietary data. For each food item, the participants accurately reported 103 responses pertaining to the frequency of consumption over the last year by checking one of nine frequency categories, which ranged from ‘almost never’ to ‘3 times/day’, followed by a question on the portion size of each food, which was divided into three categories: ‘less than one serving size’, ‘one serving size’ and ‘more than one serving size’. The food frequency was validated by two semi-quantitative food frequency questionnaires (SQFFQ1 and SQFFQ2) repeated at 1-year intervals using 3-day diet records during each of the four seasons from December 2002 to May 2004. The median spearman correlation coefficients of most food intakes, assessed repeatedly by the SQFFQ, were 0.45 for all nutrient intakes and 0.39 for nutrient densities [1]. The daily intake of each food item was converted to grams per day by multiplying the frequency per day by the portion size score [19]. The total energy, macronutrients, vitamins, and minerals were calculated from the daily consumption.

### 2.3. Identification of Dietary Patterns

The dietary pattern was measured by extracting the factors using 27 common food groups, including the food items of daily consumption, based on references [22,23], and these patterns are presented in Table 1. The factors were identified by principal component analysis rotating with an orthogonal transformation to the varimax method. The rotated factors that explained the sum of the total variance by food groups were the interpreted eigenvalues. The four most meaningful factors were determined with an eigenvalues by men (>1.5) and women (>1.6). Factor analysis was conducted using the four factors, and the results were analyzed from the derived dietary patterns as the correlation of the factors and food groups with loadings ≥ ±0.30 [24].

### 2.4. Assessment of Lung Function

Spirometry was performed by three well-trained pulmonary technologists according to the manual of the American Thoracic Society/European Respiratory Society Task Force, using the same spirometer (Vmax-229, Sensor-Medics, Yorba Linda, CA, USA) for all participants [25]. The forced expiratory volume in 1 s (FEV1), forced vital capacity (FVC), and FEV1/FVC ratio were obtained. In addition, COPD was defined as an FEV1/FVC ratio below 0.7 according to the 2018 Global Initiative for Chronic Obstructive Lung Disease guidelines [26].

### 2.5. Statistical Analysis

Principal component factor analysis was performed to identify the dietary patterns. To analyze the relationship between dietary pattern and lung function, multiple logistic regression models were adjusted for variables: Model 1: adjusted for age; Model 2: adjusted for the variable from model 1 plus smoking status (No/Yes); Model 3: adjusted for the variables from model 2 plus total energy intake (kcal/day) and body-mass index (BMI) (kg/m^2^); Model 4: adjusted for the variables from model 3 plus C-reactive protein (CRP). Quintiles of dietary intake were modeled as an ordinal variable in regression models, and the associated *p* value for the trend was obtained. Data were presented as means ± standard errors (SE) or the number of cases (percentage). Categorical data were compared using the Pearson’s chi-squared test. Continuous variables were compared using the *t*-test. All *p* values < 0.05 were considered statistically significant.

## 3. Results

### 3.1. Baseline Characteristics

The baseline characteristics of the study population are shown in Table 2. The mean age was 54.6 years, and 47.8% were men. More than half of the participants were non-smokers (68.9%). Concerning the lung function parameters, the study population had a mean FEV1 of 2.8 L (±0.7), mean FVC of 3.6 L (±0.8), and mean FEV1/FVC of 79.1% (±6.4). The mean energy and micronutrient intakes were higher in men, except for the percent of energy from carbohydrate, calcium, vitamin C, and fiber. When we analyzed the baseline characteristics according to the FEV1, the lower quartile group of FEV1 tended to be older and thinner. In addition, the lowest quartile group of FEV1 had a lower intake of total energy, as well as lower intakes of vitamin C and vitamin E, among both men and women (Appendix A).

### 3.2. Dietary Patterns among Men and Women

Table 3 shows the four major dietary patterns identified using factor analysis among men and women: Prudent pattern, involving a higher intake of potatoes, starch jelly, legumes, vegetables, kimchi, mushrooms, fruits, meat, fish and shellfish, seaweeds, soup, and other beverages; ‘Coffee, fat, and sweet’ pattern, characterized by a higher intake of sweets, oils and fats, and coffee; Westernized pattern with a higher intake of noodles and dumplings, cereals and snacks, bread, pizza, hamburger, starch jelly, nuts, fruits, meat, eggs, fish and shellfish, dairy products, and seasoning; White rice pattern with high factor loading with white rice and negative loading with whole grains.

### 3.3. Dietary Patterns and Lung Function Measurements

Table 4 and Table 5 reveal multivariable regression analyses for the association between dietary pattern and lung function adjusted for models 1–4 among men and women, respectively. Regarding the relationship with the lung function parameter of FEV1/FVC ratio, pattern 2 (coffee, fat, and sweet) showed a significantly negative association after adjusting for variables in all models (Table 4). We found no associations between the dietary patterns 1, 3, and 4 and any of the lung function parameters among men. In addition, no association was found between dietary patterns and lung function parameters among women (Table 5).

### 3.4. Dietary Patterns and COPD

Table 6 and Table 7 show the results of the multiple logistic regression models estimating the association between dietary patterns and COPD among men and women, respectively. Participants with high scores for the ‘Coffee, fat, and sweet’ pattern had a higher risk of COPD among men. This trend remained even after adjustment for the confounders of age, smoking status, total energy intake, BMI, and C-reactive protein (Table 6). In addition, when we analyzed the differences in intake based on the 27 common food groups, the COPD group showed a higher intake of sweets, kimchi, and soup, along with a relatively lower intake of starch jelly, mushrooms, fruits, and dairy products among men (Appendix A). However, there was no significant association between dietary pattern and COPD among women (Table 7).

## 4. Discussion

In this study, we evaluated the relationship between dietary patterns and lung function outcomes. We have found that a ‘coffee, fat, and sweet’ pattern, characterized by higher intake of sweets, oils, fat, and coffee, was negatively associated with lung function, particularly the FEV1/FVC ratio, and was associated with an increased prevalence of COPD in men. In addition, although the association of the ‘coffee, fat, and sweet’ pattern with FEV1 was not statistically significant, it was consistent with the FEV1/FVC ratio results. This relationship was not modified after adjustment for potential confounders, such as energy intake, age, BMI, and smoking status.

Whereas the relationship between dietary patterns and the risk of other chronic diseases such as cancer, cardiovascular disease, or diabetes, has been reported in several previous studies [8,9,10,11], relatively few epidemiological studies have reported the relationship between dietary patterns and the risk of COPD. Two prospective studies [27,28] found that the ‘prudent’ pattern, characterized by a high intake of fruits and vegetables, oily fish, and low-fat products, was negatively associated with the diagnosis of COPD. One Chinese meta-analysis involving 550,614 participants also revealed that the healthy/prudent dietary pattern was associated with a decreased risk of COPD [29]. Our findings are consistent with these studies [27,28,29], as one of the hallmarks of the ‘coffee, fat, and sweet’ dietary pattern is a low vegetable intake. Our results are also in agreement with a cross-sectional analysis from the Netherlands, which found that the ‘traditional’ pattern, represented by higher intakes of red meat, fat, coffee, and beer, was associated with a lower lung function and higher prevalence of COPD [30]. Although the pathophysiological mechanisms are unclear, the potential beneficial effects of antioxidant and anti-inflammatory nutrients in fruits and vegetables are thought to have a positive effect on lung function [31,32]. Similar to the results of previous studies [33,34,35,36,37], the group with a lower FEV1/FVC had a lower intake of vitamin C and vitamin E in our study. They also tended to eat smaller amounts of fruits, dairy products, mushrooms, and starch jelly.

Recent studies [11,38,39,40] have also demonstrated that a high intake of sweets and fat is associated with reduced lung function. Obesity has been proposed as a possible explanation for the association between a diet rich in sweets and fats and impaired lung function [41,42]. Reductions in FEV1 and FVC have been documented in extremely obese subjects. Fat accumulation in the rib cage and visceral cavity may reduce respiratory function [43]. Additionally, several studies [44,45,46,47] have reported that decreased lung function is associated with diabetes and high levels of fasting blood glucose. Diabetes and hyperglycemia may trigger oxidative stress-related inflammatory responses and alter the regulation of the inflammatory pathway [48], consequently leading to chronic lung function impairment [47]. These effects may be associated with alveolar morphological changes caused by increased oxidative stress [49]. Inflammatory changes in the pulmonary vascular tissue, along with thickening of the pulmonary capillary basement membrane, have been reported in diabetic animal models [49].

Studies on the effects of caffeine consumption on the risk of developing COPD have demonstrated inconsistent results. Hirayama et al. conducted a case control study that found a positive association between caffeine intake and the development of COPD in Japanese adults [50]. Conversely, a retrospective study has suggested that caffeine has no significant effect on the frequency of COPD exacerbations [51]. More research is needed to understand the relationship between caffeine intake and COPD.

In addition, we noted a sex-associated difference in the association of dietary pattern and lung function in this study. Dietary patterns and associated behavioral patterns may differ in men and women, and their impact on lung function could be different. There are several studies of gender differences in the effects of smoking on lung function [21,52]. Thresholds for the detrimental effects of pulmonary irritants are expected to be different in men and women [53]. Moreover, dietary macronutrients could have different effects on lung function in men and women [54]. Even we could not fully understand these various mechanisms, different approaches may be needed to analyze the associations between dietary patterns and lung function by gender.

The strengths of the current study are that it provides sufficient power analysis due to the large sample size (*n* = 5436), along with a validated and reliable food frequency questionnaire. Additionally, focusing on the overall dietary patterns rather than the individual nutrients in foods allows for a more predictive analysis of the disease risk [15,55]. Since individuals do not get nutrients from foods in isolation, we believe that our pattern analysis can provide practical guidelines for public health. Recently, dietary pattern analysis has been increasingly recognized as a complementary approach to inform public health recommendations [15,56]. The 2015 Dietary Guidelines Advisory Committee noted that the dietary patterns approach may have synergistic and cumulative effects on health and disease in its scientific report [57]. Another strength of our study is that we used spirometric measurements, which are the gold standard markers for the diagnosis of COPD.

This study has several limitations. First, our study population included only middle-aged and older adults, and the findings might not be generalizable to other age groups. A second, further limitation relates to unmeasured confounding: although variable covariates including age, smoking, total energy intake (kcal/day), BMI (kg/m^2^), and C-reactive protein were included, some potential covariates including lifestyle and sociodemographic factors may have roles as unadjusted confounders. Additionally, we could not measure the interaction [58] between dietary patterns and other covariates, such as the environment. Furthermore, although our results provide cross-sectional evidence that certain dietary patterns were associated with lung function, we did not show an association between dietary patterns and decline in lung function over a period of time. Additional longitudinal studies over an extended period of time are needed for an in-depth analysis.

## 5. Conclusions

In the current study, we observed an inverse relationship between a ‘coffee, fat, and sweet’ dietary pattern and lung function, as measured by the FEV1/FVC. Our results suggest that dietary habits may be modifiable risk factors for COPD.

## Figures and Tables

**Figure 1 nutrients-13-04348-f001:**
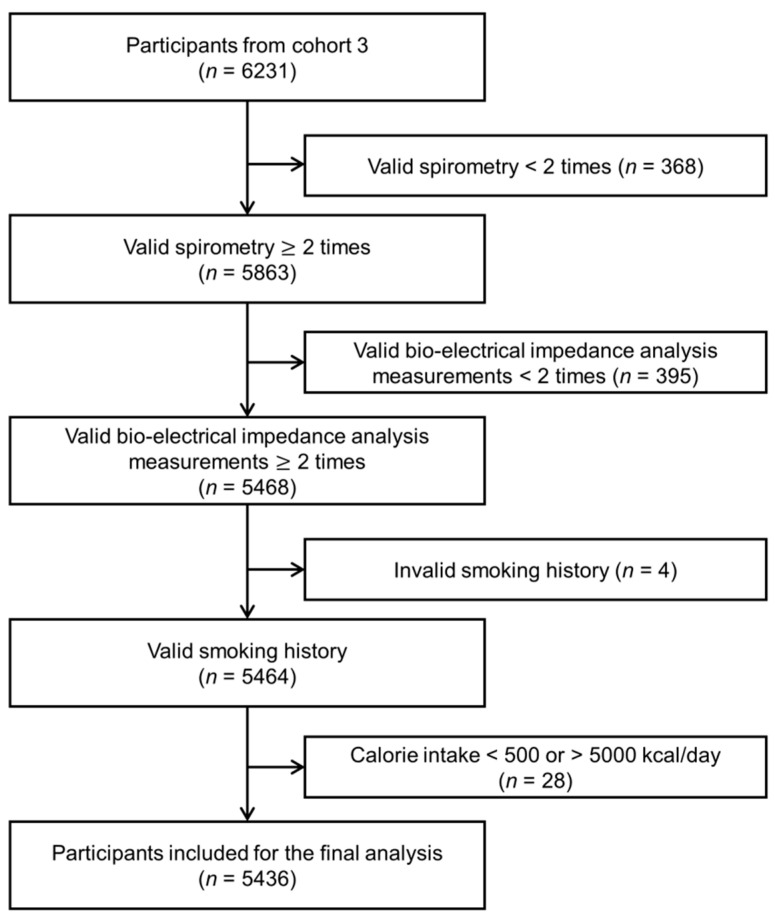
Flow diagram of study participants.

**Table 1 nutrients-13-04348-t001:** Classification of food groups.

Food Groups	Food Items
White rice	White rice
Whole grains	Cooked rice with beans, cooked rice with other cereals, cereal powder
Noodles and Dumplings	Ramyun, noodles, jajangmyeon/jjamppong, dumplings, starch vermicelli
Rice cakes	Rice cakes, rice-cake soup
Cereals and Snacks	Cornflakes, cookie/cracker/snacks
Bread	Bread, cake/chocopie
Pizza and hamburger	Pizza/hamburger
Potatoes and sweet potatoes	Sweet potatoes, potatoes
Starch jelly	Starch jelly
Sweets	Candy/chocolate, coffee sugar
Nuts	Nuts, seeds
Legumes	Beans, tofu, soy milk
Vegetables	Radish/salted radish, cabbages, spinach, lettuce, perilla leaf, salad, green vegetables, doraji/deoduck (kind of white root), bean sprouts, bracken/sweet potato stalk, red pepper leaves, leek/water dropwort, cucumber, carrot/carrot juice, onion, green pepper, pumpkin, pumpkin gruel/pumpkin juice
Kimchi	Kimchi, kkakduki/small radish Kimchi, Kimchi with liquid, other Kimchi, Korean style pickles
Mushrooms	Mushrooms
Fruits	Strawberry, muskmelon/melon, watermelon, peach, banana, persimmon, tangerine, pear, apple/apple juice, orange/orange juice, grape/grape juice, tomato/tomato juice
Meat products	Pork, beef, ham/sausage, chicken
Eggs	Eggs
Fish and shellfish	Mackerel/pacific saury/Spanish mackerel, hair tail, eel, yellow croaker, Alaska pollack, crab, clam, oyster, shrimp, dried anchovy, canned tuna, salted-fermented fish, fish paste, sushi
Seaweeds	Brown seaweed, sea mustard, dried laver
Dairy products	Milk, yoghourt, cheese, ice cream
Soups	Soybean paste/stew with soybean paste
Seasoning	Jam/honey/butter/margarine
Oils and fats	Coffee cream
Coffee	Coffee
Carbonated beverages	Carbonated beverages
Other beverages	Green tea, other drinks

**Table 2 nutrients-13-04348-t002:** Baseline characteristics of study participants.

Variables	Total (*n* = 5436)	Men (*n* = 2599)	Women (*n* = 2837)	*p*-Value
Age, years	54.6 ± 8.2	54.2 ± 8.0	55.0 ± 8.4	<0.0001
Smoking history, *n* (%)	1690 (31.1)	1656 (63.7)	34 (1.2)	<0.0001
Height, cm	160.5 ± 8.5	167.2 ± 5.7	154.3 ± 5.5	<0.0001
Weight, kg	63.4 ± 9.8	68.2 ± 9.1	58.9 ± 8.1	<0.0001
Waist circumference, cm	83.9 ± 8.5	85.0 ± 7.4	82.9 ± 9.3	<0.0001
Body mass index, kg/m^2^	24.6 ± 2.9	24.4 ± 2.7	24.8 ± 3.1	<0.0001
<18.5, *n* (%)	61 (1.1)	35 (1.4)	26 (0.9)	0.060
18.5–25, *n* (%)	3048 (56.1)	1488 (57.3)	1560 (55.0)	
≥25, *n* (%)	2327 (42.8)	1076 (41.4)	1251 (44.1)	
Fat mass index, kg/m^2^	6.6 ± 2.3	5.2 ± 1.6	7.8 ± 2.2	<0.0001
Muscle mass index, kg/m^2^	17.0 ± 1.7	18.1 ± 1.5	16.0 ± 1.2	<0.0001
Fat mass, kg	16.7 ± 5.2	14.7 ± 4.6	18.5 ± 5.1	<0.0001
Lean mass, kg	44.3 ± 8.1	50.9 ± 5.8	38.2 ± 4.1	<0.0001
hsCRP, mg/L	1.5 ± 3.3	1.6 ± 3.7	1.4 ± 2.9	0.006
Lung function parameters				
FEV1, L	2.8 ± 0.7	3.3 ± 0.6	2.4 ± 0.4	<0.0001
FVC, L	3.6 ± 0.8	4.2 ± 0.6	3.0 ± 0.5	<0.0001
FEV1, % Pred.	111.5 ± 15.7	106.5 ± 14.3	116.1 ± 15.6	<0.0001
FVC, % Pred.	104.4 ± 12.7	102.1 ± 12.1	106.5 ± 13.0	<0.0001
FEV1/FVC ratio	79.1 ± 6.4	77.0 ± 6.9	81.0 ± 5.2	<0.0001
Total energy, kcal/day	1797.8 ± 534.4	1918.0 ± 537.0	1687.7 ± 507.7	<0.0001
% of energy from carbohydrate	72.4 ± 6.5	71.3 ± 6.3	73.4 ± 6.4	<0.0001
% of energy from protein	13.0 ± 2.3	13.1 ± 2.2	12.9 ± 2.4	0.060
% of energy from fat	13.4 ± 5.1	14.4 ± 5.0	12.6 ± 5.1	<0.0001
Protein, g/day	59.0 ± 23.0	63.1 ± 22.6	55.2 ± 22.8	<0.0001
Fat, g/day	27.9 ± 16.7	31.6 ± 17.2	24.5 ± 15.4	<0.0001
Carbohydrate, g/day	322.6 ± 89.7	339.3 ± 90.6	307.4 ± 86.0	<0.0001
Calcium, mg/day	431.6 ± 242.1	425.8 ± 228.0	436.8 ± 254.3	0.090
Iron, mg/day	9.8 ± 4.3	10.0 ± 4.1	9.5 ± 4.4	<0.0001
Vitamin A, RE/day	466.9 ± 330.0	472.1 ± 307.4	462.1 ± 349.4	0.260
Sodium, mg/day	2667.2 ± 1461.2	2819.7 ± 1442.2	2527.5 ± 1464.7	<0.0001
Vitamin B1, mg/day	1.0 ± 0.4	1.1 ± 0.4	0.9 ± 0.4	<0.0001
Vitamin B2, mg/day	0.9 ± 0.4	0.9 ± 0.4	0.9 ± 0.4	<0.0001
Vitamin C, mg/day	104.0 ± 61.8	99.1 ± 57.0	108.5 ± 65.6	<0.0001
Zinc, mg/day	7.8 ± 3.1	8.3 ± 3.3	7.3 ± 2.9	<0.0001
Vitamin B6, mg/day	1.6 ± 0.6	1.6 ± 0.6	1.5 ± 0.6	<0.0001
Folate, ug/day	218.0 ± 115.6	218.2 ± 109.8	217.7 ± 120.7	0.870
Fiber, g/day	5.9 ± 2.7	5.9 ± 2.6	5.9 ± 2.7	0.370
Vitamin E, mg/day	8.1 ± 4.1	8.3 ± 3.9	7.9 ± 4.2	<0.0001
Cholesterol, mg/day	150.6 ± 115.0	162.6 ± 114.5	139.6 ± 114.4	<0.0001

Data are presented as mean ± standard error or the number of cases (percentage). Abbreviations: hsCRP, high-sensitivity C-reactive protein; FEV1, forced expiratory volume in one second; FVC, forced vital capacity; % Pred, % of predicted value.

**Table 3 nutrients-13-04348-t003:** Factor loading matrix for the major factors ascertained from the dietary patterns.

Foods or Food Groups	Pattern 1 (Prudent)	Pattern 2 (Coffee, Fat, and Sweet)	Pattern 3 (Westernized)	Pattern 4 (White Rice)
**Men**				
White rice	−0.048	0.043	0.011	0.937
Whole grains	0.088	−0.026	−0.037	−0.934
Noodles and Dumplings	0.120	0.015	0.301	0.162
Rice cakes	0.070	−0.049	0.268	−0.036
Cereals and Snacks	0.021	0.087	0.486	−0.008
Bread	−0.018	0.034	0.616	0.041
Pizza and hamburger	−0.040	0.129	0.405	0.068
Potatoes and sweet potatoes	0.356	−0.073	0.182	−0.052
Starch jelly	0.374	0.036	−0.039	−0.001
Sweets	0.062	0.917	0.066	0.018
Nuts	0.124	0.016	0.314	−0.056
Legumes	0.485	−0.015	0.105	−0.024
Vegetables	0.767	−0.026	0.086	−0.002
Kimchi	0.495	0.059	−0.205	0.065
Mushrooms	0.513	0.033	0.223	−0.083
Fruits	0.366	0.095	0.371	−0.172
Meat products	0.458	0.045	0.179	0.138
Eggs	0.198	0.016	0.332	0.000
Fish and shellfish	0.592	0.056	0.300	−0.031
Seaweeds	0.563	−0.003	0.067	−0.018
Dairy products	0.140	−0.064	0.427	−0.122
Soups	0.467	0.019	−0.087	−0.003
Seasoning	−0.081	0.0003	0.465	0.049
Oils and fats	0.026	0.915	0.067	0.019
Coffee	0.062	0.915	0.044	0.037
Carbonated beverages	0.089	0.014	0.288	0.175
Other beverages	0.302	0.032	0.164	−0.071
Variance Explained by Each Factor	3.05	2.58	2.09	1.91
**Women**				
White rice	−0.043	0.046	−0.009	0.929
Whole grains	0.041	−0.050	−0.062	−0.907
Noodles & Dumplings	0.068	0.082	0.395	0.135
Rice cakes	0.145	−0.031	0.293	0.013
Cereals and Snacks	−0.059	0.109	0.548	0.014
Bread	0.011	0.073	0.645	0.023
Pizza and hamburger	−0.060	0.047	0.427	−0.065
Potatoes and sweet potatoes	0.503	−0.037	0.045	−0.009
Starch jelly	0.146	−0.035	0.392	0.006
Sweets	0.047	0.869	0.038	0.005
Nuts	0.142	0.075	0.312	−0.049
Legumes	0.447	−0.016	0.155	−0.051
Vegetables	0.801	−0.026	0.109	0.016
Kimchi	0.520	0.081	−0.177	0.126
Mushrooms	0.474	−0.037	0.269	−0.066
Fruits	0.301	0.017	0.371	−0.185
Meat products	0.422	0.049	0.373	0.065
Eggs	0.292	0.131	0.231	−0.103
Fish and shellfish	0.602	0.041	0.338	−0.051
Seaweeds	0.630	−0.007	0.013	0.020
Dairy products	0.268	0.041	0.293	−0.089
Soups	0.469	0.030	−0.169	−0.0001
Seasoning	0.008	−0.044	0.428	0.014
Oils and fats	−0.039	0.883	0.029	0.013
Coffee	0.046	0.851	0.081	0.019
Carbonated beverages	0.029	0.132	0.125	0.069
Other beverages	0.288	0.100	0.179	−0.103
Variance Explained by Each Factor	3.16	2.36	2.33	1.81

Factors were interpreted based on the variables with a factor loading ≥ ±0.30.

**Table 4 nutrients-13-04348-t004:** Associations between dietary patterns and lung function after adjustment for different covariates among men (*n* = 2599).

	Pattern 1 (Prudent)			Pattern 2 (Coffee, Fat, and Sweet)			Pattern 3 (Westernized)			Pattern 4 (White Rice)		
	ß	95% CI	*p*	ß	95% CI	*p*	ß	95% CI	*p*	ß	95% CI	*p*
FEV1/FVC, %												
Model 1	0.0006	−0.006 to 0.007	0.860	**−0.014**	**−0.020 to −0.008**	**<0.0001**	0.007	0.001 to 0.013	0.030	−0.006	−0.012 to −0.00006	0.050
Model 2	−0.0001	−0.006 to 0.006	0.980	**−0.007**	**−0.013 to −0.0005**	**0.030**	0.005	−0.001 to 0.011	0.100	−0.005	−0.011 to 0.002	0.150
Model 3	−0.002	−0.007 to 0.003	0.450	**−0.007**	**−0.013 to −0.001**	**0.020**	0.004	−0.002 to 0.009	0.180	−0.004	−0.010 to 0.002	0.230
Model 4	−0.002	−0.007 to 0.003	0.470	**−0.007**	**−0.013 to −0.001**	**0.020**	0.004	−0.001 to 0.009	0.190	−0.004	−0.010 to 0.003	0.250
FEV1, % Pred.												
Model 1	0.0003	−0.002 to 0.003	0.850	−0.003	−0.006 to −0.0002	0.040	0.003	0.001 to 0.006	0.020	−0.0004	−0.003 to 0.002	0.780
Model 2	0.0001	−0.003 to 0.003	0.970	−0.0004	−0.003 to 0.002	0.770	0.003	0.0001 to 0.006	0.040	0.0002	−0.002 to 0.003	0.870
Model 3	−0.0008	−0.003 to 0.002	0.480	−0.0007	−0.003 to 0.002	0.580	0.002	−0.0005 to 0.004	0.120	0.0003	−0.002 to 0.003	0.830
Model 4	−0.0007	−0.003 to 0.002	0.530	−0.0007	−0.003 to 0.002	0.580	0.002	−0.0006 to 0.004	0.140	0.0005	−0.002 to 0.003	0.730
FVC, % Pred.												
Model 1	−0.0007	−0.004 to 0.003	0.680	0.001	−0.002 to 0.004	0.410	0.002	−0.001 to 0.005	0.290	0.002	−0.0008 to 0.003	0.140
Model 2	−0.0007	−0.004 to 0.002	0.660	0.002	−0.001 to 0.005	0.220	0.002	−0.002 to 0.005	0.330	0.003	−0.0006 to 0.006	0.120
Model 3	−0.0009	−0.004 to 0.002	0.530	0.002	−0.001 to 0.005	0.230	0.001	−0.002 to 0.004	0.410	0.002	−0.0008 to 0.006	0.150
Model 4	−0.0008	−0.004 to 0.002	0.570	0.002	−0.001 to 0.005	0.230	0.001	−0.002 to 0.004	0.440	0.003	−0.0007 to 0.006	0.120

Data shown are regression model ß and their 95% CI. Multivariable regression model adjusted for variables; Model 1, adjusted for age; Model 2, adjusted for variable from model 1 plus smoking status (No/Yes); Model 3, adjusted for variable from model 2 plus total energy intake (kcal/day), BMI (kg/m^2^); Model 4, adjusted for variable from model 3 plus C-reactive protein. Abbreviations: ß, coefficients; CI, confidence intervals; FEV1, forced expiratory volume in one second; FVC, forced vital capacity; % Pred, % of predicted value.

**Table 5 nutrients-13-04348-t005:** Associations between dietary patterns and lung function after adjustment for different covariates among women (*n* = 2837).

	Pattern 1 (Prudent)			Pattern 2 (Coffee, Fat, and Sweet)			Pattern 3 (Westernized)			Pattern 4 (White Rice)		
	ß	95% CI	*p*	ß	95% CI	*p*	ß	95% CI	*p*	ß	95% CI	*p*
FEV1/FVC, %												
Model 1	−0.004	−0.011 to 0.003	0.290	0.0002	−0.007 to 0.007	0.960	−0.006	−0.013 to 0.001	0.090	0.002	−0.005 to 0.010	0.540
Model 2	−0.004	−0.011 to 0.004	0.340	0.0009	−0.006 to 0.008	0.810	−0.006	−0.013 to 0.001	0.100	0.002	−0.005 to 0.010	0.520
Model 3	−0.002	−0.008 to 0.005	0.600	0.002	−0.006 to 0.009	0.650	−0.002	−0.008 to 0.004	0.450	0.0009	−0.006 to 0.008	0.800
Model 4	−0.002	−0.008 to 0.005	0.580	0.001	−0.006 to 0.009	0.700	−0.002	−0.008 to 0.004	0.450	0.0008	−0.006 to 0.008	0.840
FEV1, % Pred.												
Model 1	−0.001	−0.004 to 0.001	0.260	0.0004	−0.002 to 0.0029	0.720	−0.0002	−0.002 to 0.002	0.890	0.0008	−0.002 to 0.003	0.540
Model 2	−0.0013	−0.004 to 0.001	0.290	0.0007	−0.002 to 0.003	0.560	−0.0001	−0.002 to 0.002	0.900	0.0008	−0.002 to 0.003	0.530
Model 3	−0.0004	−0.003 to 0.002	0.700	0.001	−0.001 to 0.003	0.410	0.0005	−0.001 to 0.003	0.610	0.0005	−0.002 to 0.003	0.680
Model 4	−0.0004	−0.003 to 0.002	0.700	0.001	−0.001 to 0.004	0.360	0.0005	−0.002 to 0.003	0.630	0.0006	−0.002 to 0.003	0.630
FVC, % Pred.												
Model 1	−0.0006	−0.003 to 0.002	0.700	0.0005	−0.002 to 0.003	0.750	0.001	−0.002 to 0.004	0.400	0.0002	−0.003 to 0.003	0.910
Model 2	−0.0005	−0.003 to 0.002	0.720	0.0007	−0.002 to 0.004	0.640	0.001	−0.002 to 0.004	0.390	0.0002	−0.003 to 0.003	0.910
Model 3	0.0001	−0.002 to 0.003	0.960	0.0008	−0.002 to 0.004	0.560	0.001	−0.001 to 0.004	0.330	0.0002	−0.003 to 0.003	0.900
Model 4	0.0001	−0.002 to 0.003	0.950	0.001	−0.002 to 0.004	0.470	0.001	−0.001 to 0.004	0.340	0.0003	−0.003 to 0.003	0.830

Data shown are regression model ß and their 95% CI. Multivariable regression model adjusted for variables; Model 1, adjusted for age; Model 2, adjusted for variable from model 1 plus smoking status (No/Yes); Model 3, adjusted for variable from model 2 plus total energy intake (kcal/day), BMI (kg/m^2^); Model 4, adjusted for variable from model 3 plus C-reactive protein. Abbreviations: ß, coefficients; CI, confidence intervals; FEV1, forced expiratory volume in one second; FVC, forced vital capacity; % Pred, % of predicted value.

**Table 6 nutrients-13-04348-t006:** Odds ratios and 95% confidence intervals for the risk of COPD according to the dietary patterns among men (*n* = 2599).

	Quartiles of Dietary Pattern Score							
	Q1 (Lowest)	Q2		Q3		Q4 (Highest)		*p* for Trend
		OR	95% CI	OR	95% CI	OR	95% CI	
Pattern 1 (Prudent)								
No. of COPD case/subjects	124/649	121/650		87/650		95/650		
Model 1	1 (ref.)	1.11	0.83 to 1.50	0.83	0.61 to 1.15	0.89	0.65 to 1.22	0.290
Model 2	1 (ref.)	1.16	0.86 to 1.58	0.83	0.60 to 1.15	0.91	0.66 to 1.25	0.200
Model 3	1 (ref.)	1.14	0.83 to 1.55	0.80	0.57 to 1.12	0.85	0.59 to 1.21	0.150
Model 4	1 (ref.)	1.12	0.82 to 0.82	0.79	0.56 to 1.10	0.84	0.59 to 1.20	0.150
Pattern 2 (Coffee, Fat, and Sweet)								
No. of COPD case/subjects	93/649	104/650		119/650		111/650		
Model 1	1 (ref.)	1.30	0.94 to 1.80	1.74	1.27 to 2.40	**1.85**	**1.34 to 2.56**	**0.001**
Model 2	1 (ref.)	1.20	0.87 to 1.67	1.55	1.12 to 2.14	**1.52**	**1.09 to 2.12**	**0.030**
Model 3	1 (ref.)	1.25	0.90 to 1.74	1.59	1.15 to 2.21	**1.56**	**1.11 to 2.19**	**0.020**
Model 4	1 (ref.)	1.27	0.91 to 1.77	1.61	1.16 to 2.24	**1.57**	**1.11 to 2.21**	**0.020**
Pattern 3 (Westernized)								
No. of COPD case/subjects	145/649	102/650		90/650		90/650		
Model 1	1 (ref.)	0.83	0.62 to 1.13	0.79	0.58 to 1.07	0.84	0.62 to 1.15	0.440
Model 2	1 (ref.)	0.65	0.66 to 1.21	0.82	0.60 to 1.12	0.88	0.65 to 1.21	0.650
Model 3	1 (ref.)	0.89	0.66 to 1.21	0.81	0.59 to 1.12	0.84	0.59 to 1.20	0.620
Model 4	1 (ref.)	0.89	0.66 to 1.21	0.82	0.59 to 1.13	0.85	0.59 to 1.21	0.660
Pattern 4 (White rice)								
No. of COPD case/subjects	79/649	120/650		113/650		115/650		
Model 1	1 (ref.)	1.47	1.06 to 2.03	1.40	1.01 to 1.95	1.57	1.13 to 2.18	0.040
Model 2	1 (ref.)	1.42	1.02 to 1.98	1.32	0.94 to 1.83	1.48	1.06 to 2.06	0.100
Model 3	1 (ref.)	1.51	1.06 to 2.13	1.41	1.00 to 1.99	1.52	1.08 to 2.14	0.070
Model 4	1 (ref.)	1.48	1.05 to 2.10	1.39	0.98 to 1.96	1.50	1.07 to 2.11	0.090

Model 1, adjusted for age; Model 2, adjusted for the variable from model 1 plus smoking status (No/Yes); Model 3, adjusted for variables from model 2 plus total energy intake (kcal/day), BMI (kg/m^2^); Model 4, adjusted for variables from model 3 plus C-reactive protein. Abbreviations: COPD, chronic obstructive pulmonary disease; OR, odds ratio; 95% CI, 95% confidence interval.

**Table 7 nutrients-13-04348-t007:** Odds ratios and 95% confidence intervals for the risk of COPD according to the dietary patterns among women (*n* = 2837).

	Quartiles of Dietary Pattern Score							
	Q1 (Lowest)	Q2		Q3		Q4 (Highest)		*p* for Trend
		OR	95% CI	OR	95% CI	OR	95% CI	
Pattern 1 (Prudent)								
No. of COPD case/subjects	20/709	24/709		27/710		31/709		
Model 1	1 (ref.)	1.34	0.73 to 2.46	1.51	0.83 to 2.74	1.85	1.03 to 3.30	0.220
Model 2	1 (ref.)	1.34	0.73 to 2.47	1.51	0.83 to 2.73	1.85	1.03 to 3.30	0.220
Model 3	1 (ref.)	1.37	0.74 to 2.53	1.62	0.88 to 2.97	2.12	1.12 to 4.00	0.140
Model 4	1 (ref.)	1.37	0.75 to 2.53	1.62	0.88 to 2.97	2.12	1.12 to 4.00	0.140
Pattern 2 (Coffee, Fat, and Sweet)								
No. of COPD case/subjects	27/709	22/709		28/710		25/709		
Model 1	1 (ref.)	0.86	0.48 to 1.52	1.19	0.69 to 2.05	0.96	0.55 to 1.68	0.720
Model 2	1 (ref.)	0.85	0.47 to 1.51	1.18	0.69 to 2.04	0.95	0.54 to 1.67	0.710
Model 3	1 (ref.)	0.85	0.48 to 1.52	1.20	0.69 to 2.07	0.95	0.54 to 1.67	0.690
Model 4	1 (ref.)	0.85	0.48 to 1.52	1.20	0.69 to 2.07	0.95	0.54 to 1.67	0.700
Pattern 3 (Westernized)								
No. of COPD case/subjects	45/709	21/709		15/710		21/709		
Model 1	1 (ref.)	0.60	0.35 to 1.02	0.54	0.29 to 1.00	0.89	0.50 to 1.58	0.100
Model 2	1 (ref.)	0.60	0.35 to 1.02	0.54	0.29 to 0.99	0.89	0.50 to 1.57	0.100
Model 3	1 (ref.)	0.60	0.35 to 1.02	0.54	0.29 to 1.00	0.88	0.47 to 1.67	0.100
Model 4	1 (ref.)	0.60	0.35 to 1.02	0.53	0.29 to 1.00	0.88	0.47 to 1.67	0.100
Pattern 4 (White rice)								
No. of COPD case/subjects	21/709	26/709		34/710		21/709		
Model 1	1 (ref.)	1.02	0.56 to 1.84	1.40	0.80 to 2.45	0.88	0.47 to 1.64	0.370
Model 2	1 (ref.)	1.02	0.56 to 1.84	1.40	0.80 to 2.45	0.88	0.47 to 1.63	0.370
Model 3	1 (ref.)	1.01	0.55 to 1.88	1.39	0.77 to 2.51	0.87	0.45 to 1.70	0.370
Model 4	1 (ref.)	1.01	0.55 to 1.88	1.39	0.77 to 2.50	0.87	0.45 to 1.69	0.360

Model 1, adjusted for age; Model 2, adjusted for the variable from model 1 plus smoking status (No/Yes); Model 3, adjusted for variables from model 2 plus total energy intake (kcal/day), BMI (kg/m^2^); Model 4, adjusted for variables from model 3 plus C-reactive protein. Abbreviations: COPD, chronic obstructive pulmonary disease; OR, odds ratio; 95% CI, 95% confidence interval.

## Data Availability

This study was based on data obtained from Korean Genome and Epidemiology Study. The data that support the findings of this study are available from the corresponding author, upon reasonable request.

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
