# Peer review of "Association between Dietary Patterns and Chronic Obstructive Pulmonary Disease in Korean Adults: The Korean Genome and Epidemiology Study"

_nutrients, 2021, doi:10.3390/nu13124348_

Round 1

Reviewer 1 Report

Dear Authors,

I have read the manuscript and I send you my comments

1 methods: please add the experimental protocol

2) more data must be added on food, data on ‘coffee, fat, and sweet’ are an important limit to this study 

Author Response

Reviewer 1

I have read the manuscript and I send you my comments

1) methods: please add the experimental protocol

As pointed, we added further protocol for KoGES Ansan and Ansung study in the Methods section. (Methods 2.1)

All participants provided informed consent for the baseline data and biospecimens and underwent an interview and physical examination. The participants were questioned by trained interviewers regarding their socio-demographic status, lifestyle along with anthropometric measurement, and diet.

2) more data must be added on food, data on ‘coffee, fat, and sweet’ are an important limit to this study.

Thank you for your comments. We named pattern 2 of the four dietary patterns as ‘Coffee, fat and sweet’ pattern. However, this dietary pattern does not mean the sum of each of coffee, fat and sweet. This dietary pattern was extracted using 27 common food groups in daily consumption assessed used a semi-quantitative food frequency questionnaire (SQFFQ). The participants accurately reported 103 responses pertaining to the frequency of consumption over the last year. So this pattern represents the information of whole 27 food groups and treated as a single exposure in the analysis. We added the difference of food group intakes between normal group and COPD group in Supplementary Table 2.

Reviewer 2 Report

In this manuscript, Shin et al. studies the association between dietary patterns and chronic obstructive pulmonary disease in Korean adults comprising of 5436 patients. They used PCA and logistic regression for modeling and adjusted for confounders. The findings of the study are interesteing.
I have the following comments/questions:

1) I request the author to cite the following relevant research works as well as others in this space to highlight the importance of studies such as these and the potential for looking at cohorts to identify risk factors. 
     A) Datta A, Matlock MK, Le Dang N, Moulin T, Woeltje KF, Yanik EL, Joshua Swamidass S. 'Black Box' to 'Conversational' Machine Learning: Ondansetron Reduces Risk of Hospital-Acquired Venous Thromboembolism. IEEE J Biomed Health Inform. 2021 
        In this study, the authors look at repurposing drugs for thromboembolism in an epidemiological study and have accounted for confounding as well via temporal control. Can you explain how you have accounted for the confounders and cite this work as well as other relevant works in this space?

     B)partizan et al. Relationship between dietary patterns and COPD: a systematic review and meta-analysis, ERJ Open Res
      This paper is relevant to your study.

     C) Datta et al. “Machine Learning liver-injuring drug interactions with non-steroidal anti inflammatory drugs (NSAIDs) from a retrospective electronic health record (EHR) cohort “, PLOS Computational Biology 2021
In this work, the authors look at pairwise factors that modify the risk of the outcome. This approach may be used in your study to identify pairwise interactions that affect COPD. Please cite this work and can you explain if you looked at pairwise variables that may affect your risk of COPD?

     D)  Cho et al. Dietary patterns and pulmonary function in Korean women: Findings from the Korea National Health and Nutrition Examination Survey 2007–2011
          Another relevant work in your space that you can use to compare your results. 

2) Overall well written manuscript. However, the introduction needs more material. How is your study different than all others out there? Cite the reference mentioned in Comment 1 as well add relevant references to highlight how your study stands out.

3) “Such pattern analysis may reflect the dietary 47 habits more accurately as foods are consumed together with interaction between individ-48 ual nutrients” Can you please elaborate on this? Have you looked at interactions affecting COPD risk?

4)  “The four most meaningful factors were determined with an eigenvalue >1.3.”… How did you come up with the threshold of 1.3 to select the principal components? Have you conducted any sensitivity analyses on this ? 

5)  “We have found that a ‘Coffee, fat, and sweet’ pattern ”. It does sound you are looking at a group of variables affecting the risk of COPD. It is important to highlight how different this is from the interaction studies that you mentioned in your introduction

6) Have you looked at controlling obesity as a confounder?It appears you have used BMI as an indicator for obesity. Can you please highlight this in the manuscript?

7) Have you excluded patients form your study with a history of pulmonary disease or thrombosis? It may be relevant given that thrombosis leads to PE..

8) Have you looked at other demographic factors such as income or lifestyle in your study? 

Author Response

Reviewer 2

In this manuscript, Shin et al. studies the association between dietary patterns and chronic obstructive pulmonary disease in Korean adults comprising of 5436 patients. They used PCA and logistic regression for modeling and adjusted for confounders. The findings of the study are interesteing.
I have the following comments/questions:

1) I request the author to cite the following relevant research works as well as others in this space to highlight the importance of studies such as these and the potential for looking at cohorts to identify risk factors. 
     A) Datta A, Matlock MK, Le Dang N, Moulin T, Woeltje KF, Yanik EL, Joshua Swamidass S. 'Black Box' to 'Conversational' Machine Learning: Ondansetron Reduces Risk of Hospital-Acquired Venous Thromboembolism. IEEE J Biomed Health Inform. 2021 
        In this study, the authors look at repurposing drugs for thromboembolism in an epidemiological study and have accounted for confounding as well via temporal control. Can you explain how you have accounted for the confounders and cite this work as well as other relevant works in this space?

B) partizan et al. Relationship between dietary patterns and COPD: a systematic review and meta-analysis, ERJ Open Res
    This paper is relevant to your study.
C) Datta et al. “Machine Learning liver-injuring drug interactions with non-steroidal anti inflammatory drugs (NSAIDs) from a retrospective electronic health record (EHR) cohort “, PLOS Computational Biology 2021
In this work, the authors look at pairwise factors that modify the risk of the outcome. This approach may be used in your study to identify pairwise interactions that affect COPD.

Please cite this work and can you explain if you looked at pairwise variables that may affect your risk of COPD?

D) Cho et al. Dietary patterns and pulmonary function in Korean women: Findings from the Korea National Health and Nutrition Examination Survey 2007–2011
        Another relevant work in your space that you can use to compare your results. 

Thank you for your valuable suggestions.

We revised the manuscript with citation of recommended articles in the ‘Introduction’ and ‘Discussion’ section.

2) Overall well written manuscript. However, the introduction needs more material. How is your study different than all others out there? Cite the reference mentioned in Comment 1 as well add relevant references to highlight how your study stands out.

As pointed, we revised the manuscript with citation of recommended articles.

People eat meals with complex combinations of nutrients that are likely to be interactive or synergistic. Single nutrient analysis might be confounded by the effect of dietary. In addition, considering how foods and nutrients are consumed in combinations would reflect more closely the real world. Thus, the approach of dietary pattern analysis in which multiple dietary components are operationalized as a single exposure could provide new insights in the association of diet and COPD.

3) “Such pattern analysis may reflect the dietary habits more accurately as foods are consumed together with interaction between individual nutrients” Can you please elaborate on this?

Have you looked at interactions affecting COPD risk?

Thank you for your point. We revised the manuscript for diet pattern analysis. We intended to treat ‘diet pattern’ as a diet habit for whole food. So, we did not perform analyze the interaction between individual food. However, the interaction of food or diet pattern and other risk factors such as smoking or air pollution is an area to be investigated. We added this in the Discussion section.

People eat meals with complex combinations of nutrients that are likely to be interactive or synergistic. Single nutrient analysis might be confounded by the effect of dietary. In addition, considering how foods and nutrients are consumed in combinations would reflect more closely the real world. Thus, the approach of dietary pattern analysis in which multiple dietary components are operationalized as a single exposure could provide new insights in the association of diet and COPD.

However, there is still a lack of epidemiological studies on dietary patterns, and the impact of dietary intake on the risk of developing COPD. In this study, we aimed to determine the dietary patterns by using principal components analysis, and to investigate the association between dietary patterns and COPD. Additionally, we investigated the effect of dietary patterns on the spirometric indices.

Additionally, we could not measure the interaction between dietary patterns and other covariates such as environment.

4)  “The four most meaningful factors were determined with an eigenvalue >1.3.”… How did you come up with the threshold of 1.3 to select the principal components? Have you conducted any sensitivity analyses on this? 

To determine the number of factors in PCA, we used scree plot. We adopted the eigenvalue where the slope of the curve is clearly changed. We attached the scree plot result as below. We clarified this in the manuscript, men and women separately.

The four most meaningful factors were determined with an eigenvalue by men (>1.5) and women (>1.6)

(( Screen plot is attached to the uploaded file.))

5)  “We have found that a ‘Coffee, fat, and sweet’ pattern”. It does sound you are looking at a group of variables affecting the risk of COPD. It is important to highlight how different this is from the interaction studies that you mentioned in your introduction

We named pattern 2 of the four dietary patterns as ‘Coffee, fat and sweet’ pattern. However, this dietary pattern does not mean the sum of each of coffee, fat and sweet. This dietary pattern was extracted using 27 common food groups in daily consumption. So this pattern represents the information of whole 27 food groups and treated as a single exposure in the analysis. We added the difference of food group intakes between normal group and COPD group in Supplementary Table 2.

6) Have you looked at controlling obesity as a confounder? It appears you have used BMI as an indicator for obesity. Can you please highlight this in the manuscript?

As pointed, we adjusted BMI in the analysis. To investigate the association between dietary pattern and lung function or COPD, we analyzed the data using 4 models as below.

Model 1: adjusted for age; Model 2: adjusted for the variable from model 1 plus smoking status; Model 3: adjusted for the variables from model 2 plus total energy intake (kcal/day) and body-mass index (BMI) (kg/m²); Model 4: adjusted for the variables from model 3 plus C-reactive protein (CRP).

We described this in the ‘Methods’ section (Methods 2.5, page 4)

7) Have you excluded patients form your study with a history of pulmonary disease or thrombosis? It may be relevant given that thrombosis leads to PE.

Unfortunately, the information of pulmonary embolism or thrombosis were not included in the dataset. Also, some information of co-morbidities such as ‘pulmonary tuberculosis’ was restricted for the access.

8) Have you looked at other demographic factors such as income or lifestyle in your study? 

As pointed, we checked and adjusted the smoking in the analysis (model 2,3 and 4). Unfortunately, we did not adjust the other social factors including income, physical activity and environmental factors. We added this in the limitation.

Second, further limitation relates to unmeasured confounding: although variable covariates including age, smoking, total energy intake (kcal/day), BMI (kg/m²), and C-reactive protein were included, some potential covariates including lifestyle and sociodemographic factors may have roles as unadjusted confounders. Additionally, we could not measure the interaction between dietary patterns and other covariates such as the environment.

Round 2

Reviewer 1 Report

No comments

Author Response

There are no comments from reviewer 1.